# Mesocosm Experiments Reveal Global Warming Accelerates Macrophytes Litter Decomposition and Alters Decomposition-Related Bacteria Community Structure

**Meng Pan [1], Tao Wang [2], Bowen Hu [1], Penglan Shi [1], Jun Xu [2,\*] and Min Zhang [1,\*]**

1    Hubei Provincial Engineering Laboratory for Pond Aquaculture, Engineering Research Center of Green Development for Conventional Aquatic Biological Industry in the Yangtze River Economic Belt, Ministry of Education, College of Fisheries, Huazhong Agricultural University, Wuhan 430070, China; mt190128@163.com (M.P.); hu9599@126.com (B.H.); spl100306@163.com (P.S.)
2    Donghu Experimental Station of Lake Ecosystems, State Key Laboratory of Freshwater Ecology and Biotechnology of China, Institute of Hydrobiology, Chinese Academy of Sciences, Wuhan 430072, China; waaqgr@163.com
\*    Correspondence: xujun@ihb.ac.cn (J.X.); zhm7875@mail.hzau.edu.cn (M.Z.)

**Abstract:** Global climate change scenarios predict that lake water temperatures will increase up to 4 °C and extreme weather events, such as heat waves and large temperature fluctuations, will occur more frequently. Such changes may result in the increase of aquatic litter decomposition and on shifts in diversity and structure of bacteria communities in this period. We designed a two-month mesocosm experiment to explore how constant (+4 °C than ambient temperature) and variable (randomly +0~8 °C than ambient temperature) warming treatment will affect the submerged macrophyte litter decomposition process. Our data suggests that warming treatments may accelerate the decomposition of submerged macrophyte litter in shallow lake ecosystems, and increase the diversity of decomposition-related bacteria with community composition changed the relative abundance of Proteobacteria, especially members of Alphaproteobacteria increased while that of Firmicutes (mainly Bacillus) decreased.

**Keywords:** climate change; heat wave; litterbags; mesocosm; *Potamogeton crispus* L.





## 1. Introduction

The earth is experiencing rapid climatic changes, the mean global surface temperature will rise by 2.6 °C to 4.8 °C by the end of this century under the IPCC RCP8.5 stimulation scenario, resulting in extreme weather events at higher frequency [1]. The possibly consequent heatwaves and large temperature fluctuations will threaten global biodiversity and modify ecosystem functions, whose collapses will lead to feedbacks in global warming [2].

Aquatic macrophyte is an essential element of aquatic food web and plays an vital role in maintaining the function and biodiversity of lake ecosystem [3]. *Potamogeton crispus* is a well-known submerged aquatic macrophyte widely distributed in shallow freshwater lakes, ponds and rivers in the world [4–6]. Recent research showed that warming accelerated the growth and senescence of *P. crispus*, which shortened the time for phase shifting of the lake from a clear to turbid state [7]; the decomposition of its residue in water may further cause secondary pollution [8]. Therefore, studies in the litter decomposition process are critical for understanding the states of lacustrine ecosystems under climate change.

Decomposition is a dynamic physical and chemical change of organic matters controlled by biotic and abiotic drivers [9], such as fast leaching and slow biodegradation. The different compounds in the litter decomposed at different rates, leading to a rapid decrease in the concentration of easily decomposable nutrients and, therefore, a relative increase in which of recalcitrant compounds [10]. Litter decomposition processes are triggered and

mediated significantly by microorganisms that colonize the litter material during the degradation process [11,12]. Due to the bacteria's greater phylogenetic diversity and metabolic capacities, they could even have stronger effects on decomposition process compared to fungi [13].

Decomposition depends strongly on temperature [14] and it is sensitive to increased temperature since higher water temperature. A meta-analysis on decomposition reported that decomposition rate is expected to increase by 13.6–26.4% in response to global warming over the next 50 years [15]. Since higher temperature stimulates litter decomposition by increasing the leaching of soluble compounds and affect the abundance and structure of the decomposer communities, to directly and indirectly affect the decomposition process [16,17]. Higher temperatures promote higher rates of organic matter decomposition, which is not just a kinetic effect on bacteria reaction rates but results from litter stimulating bacteria growth [18]. Studies from aquatic ecosystems have reported overall positive effects of warming on litter decomposition, probably through a stimulation of bacteria degradation of litter with higher temperature [17,19].

Despite recent interest in the role of climate change as a driver of litter decomposition, few studies have examined the response of bacteria driven decomposition in shallow lakes to global warming and rapid changing weather events through experimental manipulation in mesocosms. For litter substrate, most studies (in terrestrial or aquatic) used terrestrial plants and focused on leaf litter. Studies demonstrated that litter decomposition rate was different between species under warming [20], few studies have considered the difference of different tissues. In order to investigate the influence of rising temperature on bacteria-directed macrophyte decomposition, we designed a short-term (2 months) mesocosm experiment and used litterbags to test the effect of constant warming and variable warming on (1) decomposition rate of stem and leaf litter of *P. crispus* over time and (2) how these changes affect the bacteria diversity and structure of litter surface in the middle stage of decomposition. We hypothesized that increased temperature would stimulate the decomposition rate of litter [21], increase the diversity of bacteria and change the community composition [20]. As some bacteria communities have capacity to recover quickly after interference [22], we think the variable warming may have obvious effects on the bacteria communities. Litter tissue can become the dominant determinant of litter decomposition under certain climate conditions [23], therefore, we assume that different tissues of submerged plant will response respond differently to climate changes.

## 2. Materials and Methods

### 2.1. Mesocosm Experiment

The mesocosms used in our experiment are situated at the Huazhong Agricultural University in Wuhan City, Central China (30°29′ N; 114°22′ E). The study experimental design have been described previously [7,24], and only a brief description is provided here. The experimental set-up included 18 fully mixed outdoor mesocosms (diameter = 1.5 m, height = 1.4 m, lake sediment: 0.1 m) to mimic shallow lake ecosystems. We simulated warming effects by a heating device (A10-3, Xin Shao Guang, China) and connected temperature sensors (DS18B20, Maxim IC, USA) connected to each tank so that we could real-time monitor the temperature in any heated and unheated treatment pair [24]. Each tank had a 10 cm-deep, well-mixed bed layer of lake sediment and was filled with 20-μm mesh filtered lake water collected from lake Liangzi to a depth of 1 m, water levels were kept constant in the tanks by resupplying tap water to compensate for evaporation losses during the experiment whenever this was not compensated for by rainfall. Before the experiment started, all tanks were left for colonization at ambient conditions for two months to stabilize the various components in the sediments. (November–December 2018). These mesocosms simulate shallow lake systems under 3 temperature scenarios with 6 replicates:(1) Control ("C", environment temperature), (2) Constant warming ("T", +4 °C above control temperature) according to IPCC climate scenario RCP8.5 [1]; (3) Variable warming ("V", we applied the fluctuations of 3–12 days' duration and 0–8 °C relative to

constant warming overall intensity to mimic a concomitant increase in extreme climatic events warming baseline as projected by relevant regional studies [25,26]).

### 2.2. Sampling and Chemical Analyses

Water samples were collected with a Plexiglas tube (diameter 70 mm; length 1 m) from each tank once a month during this study. Total nitrogen (TN) and total phosphorus (TP) were determined by spectrophotometry (UV-2800, Unico, China) after digestion with alkaline potassium persulfate [27]. Water was filtered through GF/F filters in order to determine $PO_4^{3-}$-P concentration, according to the molybdenum blue method [28], $NH_4^+$-N concentration in filtered water was determined following Nessler's reagent colorimetric method [27]. Chlorophyll-*a* was determined by water filtration on Whatman GF/C filters and spectrophotometric (UV-2800, Unico, China) analysis after ethanol extraction [29]. Dissolved oxygen (DO), pH and conductivity were measured with HACH HQD Portable Meters (HQ60d, HACH, USA).

### 2.3. Decomposition

Decomposition of macrophyte litter on the sediment surface was determined using a litter bag method [30,31]. We collected *P. crispus* from Liangzi Lake in spring for the substratum of the experiment. Leaf and stem material from these plants were separated and dried at 60 °C until constant dry weight. All of material needed for this study was collected in the same day and processed in the same period (spring 2019) to avoid the variability for this study in litter quality that could arise from collecting time. Polyester litter bags (10 × 6 cm dimensions) with 425 μm mesh size were filled with 0.50 ± 0.01 g of dry stem or leaf material, we weighted both empty bags and litter material, numbered and recorded before putting the material into the bags, analytical balance (AE224, SOPTOP, China) with accuracy of 0.1 mg was used for all weighing during this experiment. To correct for possible periphyton growth on the litterbags, we prepared the same empty bags, weighted and recorded them. We hung all these bags in the mesocosms (CS: stem litter × control; TS: stem litter × constant warming; stem litter × variable warming; CL: leaf litter × control; TL: leaf litter × constant warming; VL: leaf litter × variable warming) just above the sediment on 1 May 2019, when *P. crispus* would start to break up during this period according to our previous observation [7]). After 7, 15, 30, 45 and 60 days, two litterbags with leaf material, two with stem material and two empty litterbags were destructively sampled from each mesocosm. Half of litterbags and empty bags were dried at 60 °C until constant dry weight. The weight loss of the litterbags with plant material in them was corrected for the weight gain from the empty litterbags. Furthermore, we selected the bags (filled with stem or leaf) in both control and warming treatments of 30th day to collect bacteria.

### 2.4. DNA Extraction, Amplification and Sequencing

A cotton swab was wetted with sterile water and used to collect bacteria by wiping the entire surface of litter and then placed in 15 mL centrifuge tubes [32]. Samples for molecular analysis were kept in a freezer at −80 °C until analysis. Using the FastDNA® Spin Kit for Soil (MP Biomedicals) to extract total DNA from swabs according to the manufacturer's protocols. The barcoded primer set 338F (5′-ACTCCTACGGGAGGCAGCAG-3′) and 806R (5′-GGACTACHVGGGTWTCTAAT-3′) was used for bacteria 16S rRNA gene amplification [33]. In order to ensure the accuracy and reliability of subsequent data analysis, we first randomly selected samples for pre-experiment, to ensure that the vast majority of samples in the lowest number of cycles can amplify products with appropriate concentrations, and make full preparation for the formal test of all samples [34]. The V4-V5 region of the bacterial 16S ribosomal RNA gene was amplified by polymerase chain reaction (PCR) (95 °C for 3 min, followed by 29 cycles of 95 °C for 30 s, 55 °C for 30 s and 72 °C for 45 s, and a final extension at 72 °C for 10 min). PCR reactions were conducted in a 20 μL reaction system containing 4 μL of FastPfu Buffer (5×), 2 μL of dNTP mix (2.5 mM), 0.8 μL of

Forward Primer (5 μM), 0.8 μL of Reverse Primer (5 μM), 0.4 μL of FastPfu Polymerase and 10 ng of template DNA, 0.2 μL of BSA and add ddH$_2$O to 20 μL. Triplicate amplifications from each sample were mixed for library preparation [35].

Amplicons were extracted from 2% agarose gels and purified using the AxyPrep DNA Gel Extraction Kit (Axygen Biosciences, Union City, CA, USA) according to the manufacturer's instructions and quantified using QuantiFluor™-ST (Promega, USA). Purified amplicons were pooled in equimolar amounts and sent for paired-end sequencing on an Illumina Miseq PE300 platform (Majorbio Company in Shanghai).

## 2.5. Sequence Data Processing and Analysis

Raw reads were trimmed with Trimmomatic and Flash to remove those low quality (<20), short length (<50 bp). In brief, the raw reads were combined, denoised, trimmed, quality-filtered and aligned [36] to the SILVA 128 databases using Mothur (v1.30.2 https://www.mothur.org/wiki/Download_mothur, accessed on 2 July 2021). After initial processing, UCHIME was used for chimera removal, and operational taxonomic units (OTUs) were clustered at a 97% sequence similarity level [37] using Usearch (v7.0 http://drive5.com/uparse, accessed on 2 July 2021). To correct for the difference in sequencing depth, we normalized the sequences obtained at minimum number per sample.

## 2.6. Statistics Analysis

All statistical analyses were performed at a 0.05 statistical significance level using R software (v4.0.3; R Development Core Team), and graphs were made with "ggplot2" R package [38]. Water environment physicochemical properties including DO, pH, Conductivity, TN, NH$_4^+$-N, NO$_3^-$-N, TP, SRP and Chl.*a* between the three temperature groups were tested for differences by using generalized linear mixed models (GLMMs) with the "lme4" R package [39], and sampling date as a random effect for the models. Then, we performed post-hoc pairwise comparisons among different treatments via Tukey's test with the "lsmeans" R package [40]. Prior to the analyses, variables were transformed using log transformation as necessary. To determine the decomposition rate (*k*) in the litterbag experiment, we used the exponential decay model $M_t = M_0 \times \exp(-k_1) \times t$ where $M_t$ is the percent dry mass remaining at time (*t*) since the start of the experiment (in days) and $M_0$ is the initial dry mass, *k* is the decomposition rate in day$^{-1}$ [41]. This exponential decay model was fitted through the experimental data of leaf and stem litter for each mesocosm with the "nlsLM" function, from which the parameters *k* were derived through nonlinear algorithms [30]. Next, the decomposition rate (*k*) was analyzed using generalized linear models (GLMs) after log transformation, and post-hoc pairwise comparisons via Tukey's test was also performed with the "lsmeans" R package. Principal coordinate analysis (PCoAs) based on Bary-curtis distances were performed with the "vegen" R package. Next, permutational multivariate analysis of variance (PERMANOVA) with 999 permutations was carried out on Bary-curtis distances in order to explore whether variations in community composition can be explained by the treatments; this analysis was performed using the function "Adonis" from the "vegen" R package. Alpha diversity was calculated using Mothur (v1.30.2). Further, we tested treatment effects on the most abundant phyla using GLMs methods and Tukey's test. Taxon abundance data were logit transformation ($\log[(y + \varepsilon)/(1 - y + \varepsilon)]$, $\varepsilon$ was the minimum non-zero data y) [42] to improve normality.

## 3. Results

### 3.1. Conditions in the Experimental Mesocosms

During the experiment, the temperature of treatments showed an upward trend (Figure 1). Water temperatures were continuously kept at +4 °C higher in the constant warming treatment (T) relative to the control treatment (C); in the variable warming treatment, the temperature fluctuated relative to T. The temperature control at the expected range indicating that experimental warming was successful overall. As shown in Table 1, although some environmental factors were not significantly different in the treatments

compared to C, such as TP, $NO_3^-$-N, Chl. *a* ($p > 0.05$, Tukey's test), other environmental factors demonstrated variations in the treatments, such as conductivity and $NH_4^+$-N significantly higher in T and V than in C ($p < 0.05$, Tukey's test); DO, pH and TN in T significantly higher than in C, and $PO_4^-$-P higher in V than in C ($p < 0.05$, Tukey's test).

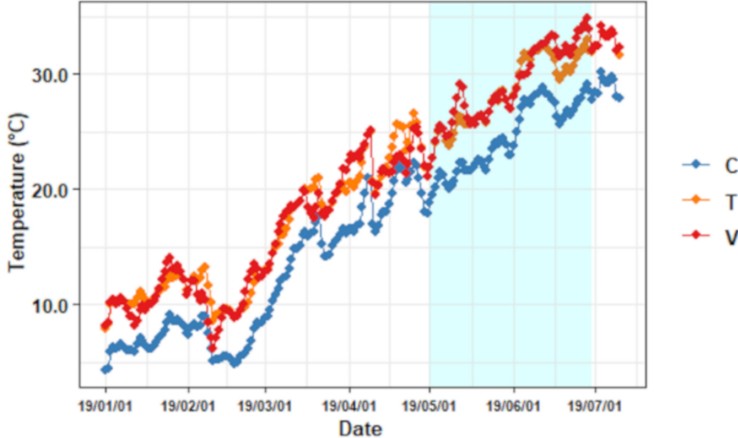

**Figure 1.** Temperature trend for all the treatments during the experiment (indicated by blue shading). C, ambient temperature; T, +4 °C constant warming; V, variable warming.

**Table 1.** Mean water chemistry from mesocosms under the C, T, V treatments. Values were calculated from monthly measurements of six replicate mesocosms in May and June 2019. Values are means ± se. C, ambient temperature; T, +4 °C constant warming; V, variable warming. Significant ($p < 0.05$) differences among treatments are indicated by letters (a, b) above the data.

| Variable | Treatment | | |
| :---: | :---: | :---: | :---: |
| | **C** | **T** | **V** |
| DO ($mg·L^{-1}$) | $3.03 \pm 0.73$ [b] | $6.05 \pm 0.70$ [a] | $3.85 \pm 0.52$ [b] |
| pH | $7.54 \pm 0.17$ [b] | $8.44 \pm 0.22$ [a] | $7.99 \pm 0.21$ [a,b] |
| TN ($mg·L^{-1}$) | $0.64 \pm 0.08$ [b] | $1.04 \pm 0.11$ [a] | $0.82 \pm 0.05$ [a,b] |
| TP ($mg·L^{-1}$) | $0.10 \pm 0.02$ [a] | $0.17 \pm 0.05$ [a] | $0.14 \pm 0.03$ [a] |
| Chl.*a* ($μg·L^{-1}$) | $6.07 \pm 1.51$ [a] | $15.10 \pm 5.13$ [a] | $3.71 \pm 0.65$ [a] |
| Conductivity ($μS·cm^{-1}$) | $177.33 \pm 9.09$ [b] | $247.08 \pm 15.73$ [a] | $248.61 \pm 7.31$ [a] |
| NH4$^+$-N ($mg·L^{-1}$) | $0.10 \pm 0.03$ [b] | $0.22 \pm 0.06$ [a] | $0.18 \pm 0.030$ [a] |
| NO3$^-$-N ($mg·L^{-1}$) | $0.11 \pm 0.01$ [a] | $0.15 \pm 0.02$ [a] | $0.13 \pm 0.02$ [a] |
| SRP ($μg·L^{-1}$) | $27.33 \pm 8.24$ [b] | $67.47 \pm 13.23$ [a,b] | $118.03 \pm 34.15$ [a] |

*3.2. Litter Decomposition Rates*

After 60 days, 90.61% ± 0.07, 98.41% ± 0.05, 93.49% ± 0.05 of litter were lost in CS (stem × control), TS (stem × constant warming), VS (stem × variable warming) litterbags and 86.74% ± 0.11, 94.87% ± 0.02, 96.34% ± 0.03 in CL (leaf × control), TL (leaf × constant warming), VL (leaf × variable warming) litterbags (mean ± se). Warming treatments (T and V) showed a positive effect on litter loss during the early stage (0–15 days) of decomposition, and this effect gradually weakened over the decomposition time (Figure 2). Dry mass remaining of leaf and stem litter decreased continuously over time and during 15–30 days, dry mass remaining slowed down, after 30 days, the decomposition rate of litter in the two warming treatments almost approached zero, while the decomposition of litter in the control group did not stop completely at the end of the experiment (Figure 2). Based on the Olson exponential decal model, the biomass loss was translated into the decomposition rate (*k*, day$^{-1}$, Figure 3), which varied among different temperature treatments and it was not affected by litter tissue type, variable warming showed a greater promoting effect, especially for leaf litter (Figure 3, Table S1).

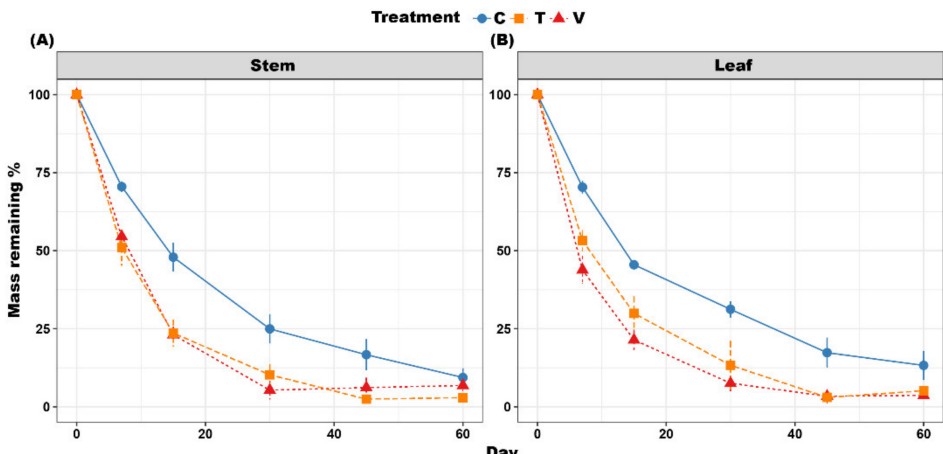

**Figure 2.** Mass remaining of Stem (**A**) and Leaf (**B**) over 60 days in C, T and V treatments. C, ambient temperature; T, +4 °C constant warming; V, variable warming. All data are presented as mean ± se.

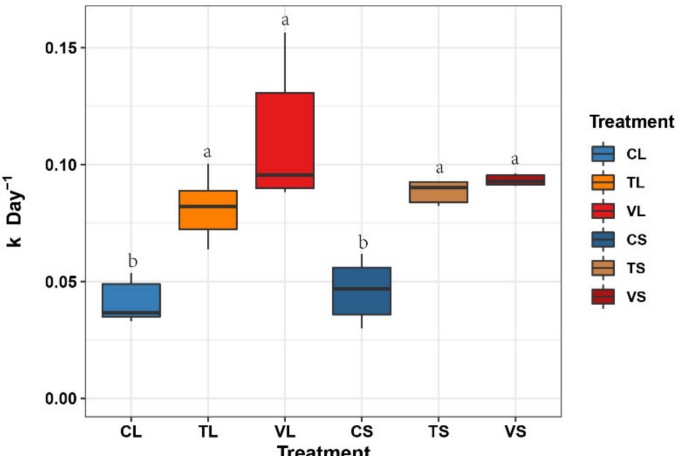

**Figure 3.** Comparison of litter decomposition rate ($k$, Day$^{-1}$) in Stem and Leaf among different temperature treatments. Treatments with different lowercase letters indicate significant ($p < 0.05$) differences between treatments. C, ambient temperature; T, +4 °C constant warming; V, variable warming; L, leaf litter; S, stem litter. All data are presented as mean ± se. Significant ($p < 0.05$) differences among treatments are indicated by letters above the bars.

### *3.3. Bacterial Alpha Diversity*

The richness of bacterial communities was estimated by the Chao index, and the diversity of these communities were determining using the Shannon index (Figure 4). There was a higher bacteria diversity of stem and leaf litter in the treatments relative to the control, and this trend is especially obvious under variable warming (V). Neither Tissue type nor constant warming affected the richness and diversity of bacterial significantly while variable warming significantly increased the richness of bacterial (both stem and leaf) and diversity of bacteria (stem).

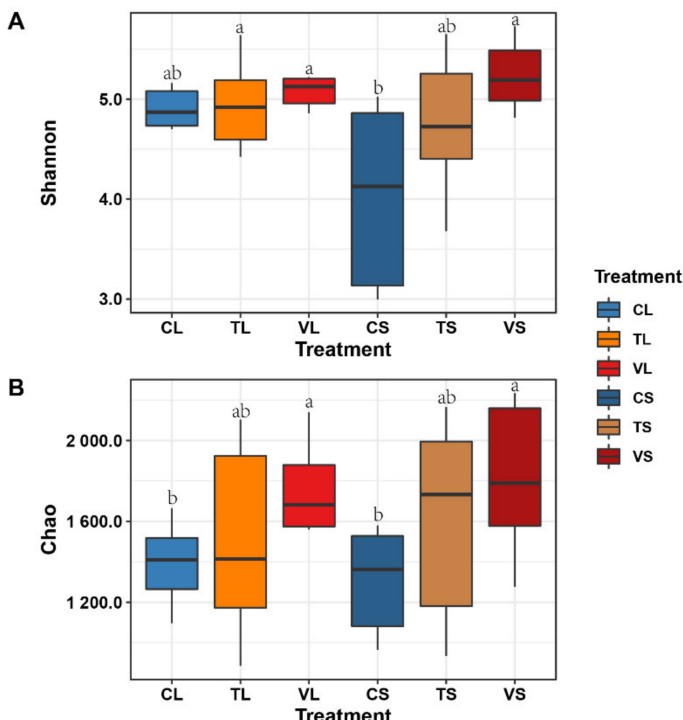

**Figure 4.** Differences of bacteria diversity among treatments in different seasons. (**A**) Shannon index of stem bacteria; (**B**) Chao index of stem bacteria; C, ambient temperature; T, +4 °C constant warming; V, variable warming; L, leaf; S, stem All data are presented as mean ± se. Significant ($p < 0.05$) differences among treatments are indicated by letters above the bars.

### 3.4. Bacterial Community Composition

PCoA analysis revealed a clear grouping of bacterial community experiencing either ambient temperature (C) and experimental warming treatments (T and V) (Figure 5), indicating that bacterial community structure was significantly impacted by warming treatments (PERMANOVA, Stem: $p = 0.035$; Leaf: $p = 0.008$).

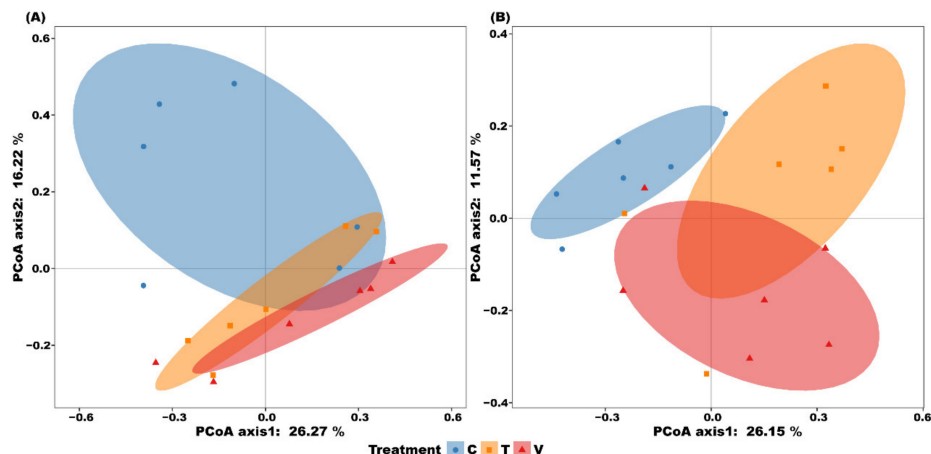

**Figure 5.** PCoA plots derived from the Bray-Curtis dissimilarities of bacterial community composition for Stem (**A**) and Leaf (**B**) litter. C, ambient temperature; T, +4 °C constant warming; V, variable warming.

Through multivariate analysis of variance, except Bacteriodota and Patescibacteria, the relative abundance of dominant bacteria phylum (>1%) was not affected by litter tissue types (Figure S1, Table S2). At different taxonomic levels, we focused on the variation

of some dominant units (Figure 6). The most predominant phyla in all the samples were *Proteobacteria* (39% ± 0.17), *Firmicutes* (21% ± 0.20), *Bacteroidetes* (11% ± 0.10) and *Actinobacteria* (7% ± 0.05). We found that compare with C, the relative abundance of *Proteobacteria* ($p = 0.03$) significantly increased in the TL, while the relative abundance of *Firmicutes* significantly decreased in the TL ($p < 0.01$) and VS ($p = 0.04$). In addition, the relative abundances of *Chloroflexi* significantly increased in VL ($p = 0.01$), TL ($p = 0.01$) and VS ($p = 0.01$). In general, 2, 7, 4 and 2 dominant phyla with significant changes were observed in VL, TL, VS and TS. Within these phyla, further investigation at class level showed that *Alphaproteobacteria* significantly increased in the VL ($p = 0.05$) and TL ($p = 0.03$) while *Bacilli* significantly decreased in the TL ($p = 0.047$), VS ($p < 0.01$) and TS ($p < 0.01$).

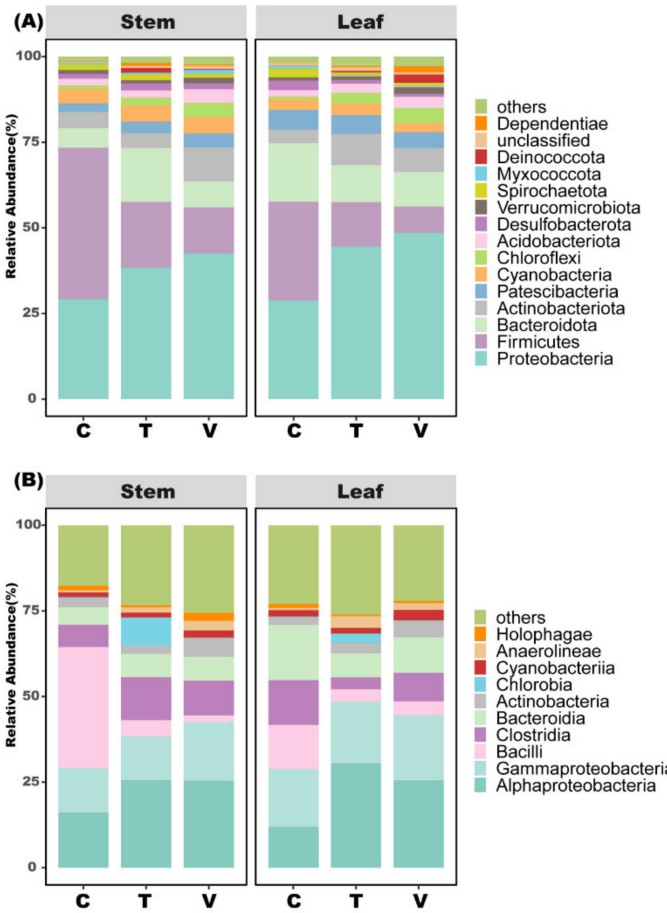

**Figure 6.** The relative abundance of the phylum (**A**) and class (**B**) taxa of bacteria associated with stem and leaf litter decomposition. The presented relative abundances are based on the 97% similarity clusters of the OTUs. C, ambient temperature; T, +4 °C constant warming; V, variable warming.

## 4. Discussion

Global warming is predicted to have strong impacts on the decomposition process in aquatic ecosystems [30]. In our study, constant warming and variable warming stimulated the decomposition rate, which has been widely reported in aquatic [30] and terrestrial ecosystems, and thus seem to be a common response to warming across ecosystems [43]. Therefore, our first hypothesis was confirmed, constant and variable warming stimulate the decomposition of aquatic macrophyte litter, especially in the early stage. Our previous research has shown that global warming will increase the growth and senescence of *P. crispus*, which will lead lakes to earlier turbid-stage in summer [7]. In addition, our study may accurately reflect the accelerated decomposition of *P. crispus* after its earlier senescence, indicating that lake ecosystems will be heavily affected in a short time for a large amount of nutrients due to fast decomposition under global warming.

There was no significant difference in decomposition rate and bacteria diversity between stem and leaf litter. A recent study showed that warming and leaf litter functional diversity, not litter quality, drive decomposition in a freshwater ecosystem [44]. A meta-analysis suggest that litter type on decomposition, at least in shallow lakes dominated by submerged plants, are less significant than that of increased temperature [45]. Little is known on how decomposition will respond to increased temperature along with shifts in different macrophyte tissue. Different macrophyte litter tissue may have no effect on decomposition.

At the early stage of decomposition, both constant and variable warming increased the decomposition rate significantly, this may be result for that higher water temperatures can accelerate litter mass loss directly by promoting leaching of soluble compounds [46] or indirectly by enhancing microbial degradation [17]. The influence of increased temperature became continuously weaker over decomposition period, and litter decomposition rate slowdown in two warming treatments, this may be due to the presence of the refractory components in litter [46]. At the end of decomposition, the dry mass remaining under the three treatments tended to be the same, indicating that warming did not significantly increase the total decomposition amount.

Variable warming showed a greater stimulation on litter decomposition and significantly affected the bacteria diversity of litter while constant warming just stimulated the decomposition rate and insignificantly affected the bacteria diversity.

Warming treatments stimulate the decomposition rate of litter, which may leads to a higher bacterial richness [13], and may have played a promoting role in the early stage of litter decomposition, easily decomposed substances in the warming treatments were decomposed earlier. However, since the warming groups has already entered the stage of decomposing refractory compounds in advance, we cannot predict the influence of increasing bacterial richness on the subsequent decomposition rate from the perspective of decomposition rate. Therefore, in future studies, we suggest measuring the bacterial community in the complete time series for further discussion.

The inconsistent effect of constant warming and heatwave on litter bacterial may be due to a consensus conclusions drawn in a varies of studies examining disturbance: the microbial communities are not resistant to many factors but highly resilient to disturbance [22,47], Finlay indicated that the composition of the microbial community usually changes when environmental factors change at high amplitude or high frequency. For example, in similar mesocosm experiments, the researchers found that heating induced a positive effect on the bacteria within a few days [48] and long-term warming had not significant effects on bacterial alpha diversity but significantly shifted bacterial community composition [49]. In addition, some studies reported the increase in bacterial diversity during the late stage of organic matter decomposition [50,51]. Therefore, the stimulation of warming on bacterial diversity found in this study is likely to be indirectly caused by accelerated decomposition.

Temperature is an important factor affecting the structure of microbial community [52], a short-term laboratory incubation experiment had confirmed that changes in microbial community composition may affect the decomposition of dissolved organic matter leachates from plant litter [53]. The analysis of bacterial community was mainly based on $\alpha$ diversity (Shannon and Chao index) and $\beta$ diversity (community structure and community composition), from these two aspects, the changes of bacterial diversity and community composition were not completely consistent, it's not a contradiction, according to our investigation, this change is mainly caused by some dominant bacteria involved in the decomposition process. Our data demonstrated that the abundance of dominant phyla *Proteobacteria*, *Firmicutes* and *Chloroflexi* changed significantly under the stimulation of temperature during the process of decomposition and their functions in plant residue decomposition have been evidenced [54]. Many studies have shown that *Alphaproteobacteria* are the center of the nitrogen cycle, taxa in *Rhizobiales* are able to link iron reduction and denitrification to photosynthesis, their role in lignin decomposition likely involves their ability

to fix nitrogen [55–57]. *Rhizobiales* can synthesize and accumulate poly-3-hydroxybutyrate (PHB) in the cell under carbon-rich conditions and utilize it under carbon-limited conditions [58], which might be the main reason for its increase throughout the decomposition under warming treatments with limited available organic carbon. The rising temperature may change the physiochemical parameters of water body by stimulating the decomposition of litter, and may have an indirect effect on microorganisms [19]. Warming treatments (T and V) in litter considerably enhanced the relative abundance of genera *Rhizobiales*, which may help to improve the utilization rate of nitrogen and alleviate the nitrogen restriction during decomposition [59], thus promoting the metabolism of other decomposers, this was also evidenced by the increase of TN and $NH_4^-$-N concentrations in T and V groups. Some studies have shown that the relative abundance of *Alphaproteobacteria* increased in later stage of decomposition [60], this may be due to their ability of decomposing refractory organic matter [61] and it has been reported that the high content of refractory organic matter (such as cellulose and lignin) can gather these units [56].

The relative abundance of *Firmicutes* was significantly increased under warming treatments, with the decreased substrate. *Bacillales*, which account for a large proportion in *Firmicutes*, are ubiquitous, endospore-forming, gram-positive bacteria that are of high economic importance due to their specific characteristics, such as their ability to colonize plants; to produce extracellular cellulase, biofilms and antibiotics; and to induce the synthesis of plant hormones [62]. The rapid loss of dry mass under warming treatments may weaken the colonization ability of *Bacillales*, thus leading to a decreasing of their relative abundance. We believed that it was the different substrates that *Firmicutes* and *Chloroflexi* preferred to which leading the opposite change of their abundance under warming treatments [51,58,63].

*Chloroflexi* are commonly found in soil or sediment communities [64], our results confirm the finding of another study that higher *Chloroflexi* abundance to be related with increased temperature [47]. Moorhead and Sinsabaugh (2006) have functionally partitioned diverse microbial communities into three decomposition guilds [65], according to this terminology, another study suggested that *Firmicutes* could be qualified as decomposers breaking down fresh organic matter and *Chloroflexi* as miners devoted to the humified organic matter [43,66].

The effect of warming on the bacteria involved in decomposition were mainly indirectly caused by increasing rate of litter decomposition, and in this process, the bacteria community composition changed. we still need more studies to consider the effects of global warming on decomposition in a broader community of decomposers and their interactions (e.g., bacteria, fungi and invertebrates). In addition, we only have one point of sampling at the time period (7 to 30 days) when greatest mass loss was happened with the litterbags, we might have missed the bacterial community responsible for the accelerated decomposition and limited our insights into any shifts in bacterial community composition. We believe that in future studies, it is also crucial to study the temporal variation of decomposing bacterial communities during litter decomposition under climate change.

**Supplementary Materials:** The following are available online at https://www.mdpi.com/article/10.3390/w13141940/s1, Figure S1: The scatterplot of decomposition rates of (A) Stem and (B) Leaf litter in each tank (C, T, V with 6 replicates). The 1.0 on the vertical axis represent 100%, Figure S2: Relative abundances of the dominant bacterial phyla (>0.01) under different warming conditions. All data are presented as mean + se, Table S1: Results from Tukey's post hoc comparing the differences of k of different treatments for in stem or leaf litter respectively. * $p < 0.05$; ** $p < 0.01$. C, ambient temperature; T, +4 °C constant warming; V, fluctuate warming, Table S2: Results from GLM comparing the differences of the relative abundance of different treatments for decomposition-related Phylum (>1%) in stem or leaf litter respectively. * $p < 0.05$; ** $p < 0.01$; *** $p < 0.001$. C, ambient temperature; T, +4 °C constant warming; V, fluctuate warming.

**Author Contributions:** J.X. and M.Z. initiated and designed the research; M.P., T.W., B.H. and P.S. performed the research and collected samples; M.P. analyzed microbial diversity; T.W. performed the statistical analysis; M.P. wrote the manuscript, J.X. and M.Z. edited the manuscript. All authors have read and agreed to the published version of the manuscript.

**Funding:** The authors are grateful to the financial support from the Water Pollution Control and Management Project of China (Grant No. 2018ZX07208005), the National Key R and D Program of China (Grant No. 2018YFD0900904 and 2019YFD0900601) and the International Cooperation Project of the Chinese Academy of Sciences (Grant No.152342KYSB20190025).

**Data Availability Statement:** Sequences are published in the NCBI Raw Sequence Read Archive (SRA), under the study accession number PRJNA705584.

**Acknowledgments:** We thank Jinyu Huang for constructing and maintaining the temperature regulatory system. In addition, we thank Huan Zhang and Peiyu Zhang for suggestions during the experiment. We also thank Mingjun Feng and Haowu Cheng for their help during the experiment.

**Conflicts of Interest:** The authors declare that they have no known competing financial interests or personal relationships that could have appeared to influence the work reported in this paper.

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
