# Peer review of "Mesocosm Experiments Reveal Global Warming Accelerates Macrophytes Litter Decomposition and Alters Decomposition-Related Bacteria Community Structure"

_water, doi:10.3390/w13141940_

Round 1

Reviewer 1 Report

The article entitled: Mesocosm Experiments Reveal Global Warming Accelerates Macrophytes Litter Decomposition and Alters Decomposition-Related Bacteria Community Structure wrote by Meng Pan, Tao Wang, Bowen Hu, Penglan Shi, Jun Xu,and Min Zhang is interesting and I accept it for publication in Water after minor revisions. I suggest add some information to Discussion section about influence of temperature on physicochemical parameters of water. My minor comment I give in a text of this paper. I think the article will be better if authors modernized the text according my comments. I accept this paper after minor revision.

Author Response

 (The paper also makes the corresponding marks, changes made to the manuscript are indicated by the corresponding line numbers “L”).

Reviewer: I suggest add some information to Discussion section about influence of temperature on physicochemical parameters of water. My minor comment I give in a text of this paper. I think the article will be better if authors modernized the text according my comments. I accept this paper after minor revision.

Accepted as follows:

  1. Potamogeton crispus is a well-known submerged aquatic macrophyte widely distributed in shallow freshwater lakes, ponds and rivers in the world [1-3]. (L37)
  2. Decomposition is a dynamic physical and chemical change of organic matters controlled by biotic and abiotic drivers [4], such as fast leaching and slow biodegradation. (L43 – L44)
  3. The aim of this study is to understand how climate warming will influence the decomposition rate of macrophyte in shallow lakes and how decomposition-related bacterial will change. (L80 – L82)
  4. All statistical analyses were performed at a 0.05 statistical significance level using R software (ver. 4.0.3; R Development Core Team), and graphs were made with “ggplot2” R package [5]. (L173)
  5. Significant (P<0.05) differences among treatments are indicated by letters (a, b, c) above the data. (L261, L272 – L273 and L281 – L282).
  6. There was no significant difference in decomposition rate and bacteria diversity between stem and leaf litter. (L303 – L304)
  7. Different macrophyte litter tissue may have no effect on decomposition. (L309 – L310)
  8. At the end of decomposition, the dry mass remaining under the three treatments tended to be the same, indicating that warming did not significantly increase the total decomposition amount. (L317 – L319)
  9. Variable warming showed a greater stimulation on litter decomposition and significantly affected the bacteria diversity of litter while constant warming just stimulated the decomposition rate and insignificantly affected the bacteria diversity. (L320 – L322)
  10. The rising temperature may change the physiochemical parameters of water body by stimulating the decomposition of litter, and may have an indirect effect on microorganisms [6]. (L365 – L367)
  11. The relative abundance of Firmicutes was significantly increased under warming treatments, with the decreased substrate. (L376 – L377)
  12. The effect of warming on the bacteria involved in decomposition were mainly indirectly caused by increasing rate of litter decomposition, and in this process, the bacteria community composition changed. (L393 – L395)

Reviewer 2 Report

This is an interesting study asking to what effect might a constant vs non-constant heating of replicated shallow lake mesocosms lead to measurable differences in P. crispus decomposition rates and the associated microbial community. While the results generally appear to support the claims made by the authors, there are some methodological gaps that are never mentioned in the Methods or Discussion. Furthermore, the writing requires some heavy editing for greater clarity. Below, I've assembled some examples of where and how the writing could be clarified.

No mention is made of how the 0.5 g samples of leaf and stem tissues were handled before being placed in the litter bags. In my experience vegetation that's been dried to 60 C can be quite brittle. Could it be that drying the P. crispus artificially enhanced its decomposition beyond what might have been expected if that material had not been oven-dried first?

For a second example, there should be mention in the Discussion of the limited insights into any shifts in bacterial community composition because this team only sampled the community at 1 point in time. Unfortunately, from my perspective, this 1 point in time also happened to be well past the point of greatest mass loss associated with the litterbags (7 and 15 days). Your litter bag extractions might have missed the bacterial community responsible for the accelerated decomposition by sampling 30 days into the experiment. Furthermore, no mention is made of whether the "control" empty litter bags were extracted for 16S rRNA analyses; this should be noted explicitly to clarify "background" microbial community composition separate from that of the two tissue types.

In Section 3.4, the authors should explain what HS and HL represent; I suspect typos. Furthermore, why are there 2 sets of p values in this paragraph? I do not believe p values magically delineate significance at p<=0.05, but how should line 239 be interpreted when the authors report a result like "p=0.047, post hoc Tukey's test p=0.078"? 

In Section 4 (Discussion), you've argued "Here we show that different macrophyte litter tissue may have no effect on decomposition." This is only true if one were to skip Fig. 3, which shows (to me) that there might be an interaction between litter type and increased temps. I understand the authors selected a LMME, so why not explore whether temperature, tissue type, and their interaction were statistically significant? Why am I interested in the potential for a temperature X tissue type interaction? Because Fig. 3 also shows that median decomp rates are somewhat greater for leaf tissue than for stem tissue under V treatments, even though for control treatments, leaf median decomp rates were less than those of stem tissue. 

Your readers would be helped if you could clarify how you think about alpha diversity. You could mention, for example, that your 2 measures of alpha diversity you calculated were Chao and Shannon, which treat the data slightly differently: Chao assumes X, whereas Shannon assumes Y. Please do not write "Variable warming produced a significant positive effect on diversity and richness of stem litter whereas significantly 
 increased the richness of leaf litter" (lines 221-2) because this then implies diversity and richness are not the same thing. But you seem to treat
 them the same.

On line 298 you have written: "We observed that variable warming showed a greater stimulation on litter decomposition and significantly affected 
the bacteria diversity of litter while constant warming just stimulated the decomposition rate and insignificantly affected the bacteria diversity."

This is not true for stem litter (see Fig 3A) as indexed by k... and it's only true related to bacteria if you look only at Chao and Shannon diversity indices and not the Bray-Curtis dissimilarity PCAs... And I cannot agree with 
this notion that you can see a "greater stimulation on litter decomposition" than constant warming in Figure 2. Speaking of Figure 2, are those SE bars? How is a reader to know this?

As a reviewer, this sentence starting line 300 was emblematic of the unclear writing that must be improved. Please note this is *not* the only instance where editing is required to improve the readability of the manuscript. "This result may indicated that the bacteria richness may be associated with decomposition, as they are all affected by environment, but the increased bacteria richness in litter could not predict the rate of litter decomposition as demonstrated by a transplant experiment [11] that warming can increase the growth rate of bacterial directly and faster decomposition may lead to a large release of nutrients from litter, indirectly leading to higher bacteria abundance."

First, the mention of ref. 11 strikes me as odd. If that study was meant to support your findings, be more clear about which results in particular. Was it that variable and constant warming appeared to the authors (not to this reviewer!) to have different effects on decomposition rates and microbial community characteristics? If so, please state this more directly. Second, what does "they are all affected by environment" mean? Does it mean that all bacteria are affected by environmental factors such as temperature? If so, please state this. Third, the mention of predicting decomposition rates reminded me the authors never presented a scatterplot of decomposition rates (however you calculate them; see below) against bacteria abundance or diversity. Why not?

The overall work as presented has a number of loose ends. For example, on line 51 the authors wrote "A meta-analysis on decomposition reported that decomposition rate is expected to increase by 13.6-26.4% in response to global warming over the next 50 years [13]." That's interesting to me. So why not then report then how your constant warming of 4C increased decomp over 7 days, 15 days, 30 days? And then variable warming? Was it within the range state or smaller or greater. 

I'm unclear how Moorhead and Sinsabaugh's "3 decomposition guilds" (ref 63) might have held up to the test of time since that work is almost 15 years old.

Please consider plotting temp effects and tissue effects on the top 20 most abundant phyla or classes as done on slide 5 here
http://www.evolution.unibas.ch/walser/bacteria_community_analysis/2015-02-10_MBM_tutorial_combined.pdf

Figures and tables

Fig 2: This is the only figure where the legend reads CVT. Recommend for consistency presenting as CTV.

I'm a bit confused on referring text for Fig. 2 that claims "Warming treatments (T and V) showed [statistically] significant positive effects on litter loss during the early stages (0-15 d) of decomposition" bc this implies that T&V did NOT have statistically significant effects after 15 d. The text should be more explicit about when treatment effects showed statistically significant differences. 

Fig 3: If your k were calculated over the full 60-day experiment, why not calculate and present (supplemental info?) the k's for shorter intervals such as 7 and 15d? Please present statistical comparisons between tissue type as well to support text reference claiming no diff in decomp rates between tissue types.

Fig 4: Please make y axes consistent for the pairs of diversity indices shown (right now your stem Shannon plot runs 2.5-6, whereas your leaf Shannon plot runs 3.6-5.6). You should not expect your readers to have to adjust the plots in their mind to account for such axis differences.

Some mention should be made of the differences in climate projections for air temperatures vs shallow lake systems, given the high specific heat capacity of water. This fact might also help explain why your variable temperatures did not seem to have as pronounced an effect on either decomposition rates or the microbial community.

Personally, I see no need to report % to the nearest 0.01%. I know it might be for consistency sake, but I believe it dilutes the impact of your writing to write 21.04%+/-0.20.

Let's think a bit about Fig 6: All 6 reps per treatment are shown in Fig 5. I strongly recommend you report the error bars as cleanly as possible for each of the 6 reps that were *averaged*(?) to obtain these relative abundances in Figure 6. Or simply plot 18 bars (all C, all T, all V) per tissue type? Without these error bars (or some index of within-treatment variances), it is impossible to  know if the bacterial communities (as indexed by relative abundances of phyla/classes) differ. My prediction is that the within-treatment variances will be far greater than the between-treatment variances as shown. Convince me (the reader) you are right and I am wrong! Until I understand within-treatment variances in community profiles, I am somewhat skeptical of the importance and significance of the differences claimed. And not just because the community was only profiled once over the course of the 60-day experiment. Mostly because aside from the interquartile ranges and whiskers from Fig 4 and the PCoA replicate datapoints in Fig. 5, no variances are shown related to dominant phyla/class abundances.

This is strong work. I hope these types of comments are helpful as you get this story out. 
